# Physics-informed neural networks integrating compartmental model for analyzing COVID-19 transmission dynamics

Xiao Ning
State Key Laboratory of
Bioelectronics, School of Biological
Science and Medical Engineering,
Southeast University
Nanjing, P.R. China
ningxiao@seu.edu.cn

Yongyue Wei
Public Health and Epidemic
Preparedness and Response Center,
Peking University
Beijing, P.R. China
ywei@pku.edu.cn

Feng Chen*
Center for Global Health,
Departments of Epidemiology and
Biostatistics, Nanjing Medical
University
Nanjing, P.R. China
fengchen@njmu.edu.cn

## ABSTRACT

Modelling and predicting the behaviour of infectious diseases is essential for early warning and evaluating the most effective interventions to prevent significant harm. Compartmental models produce a system of ordinary differential equations (ODEs) that are renowned for simulating the transmission dynamics of infectious diseases. However, the parameters in compartmental models are often unknown, and they can even change over time in the real world, making them difficult to determine. This paper proposes an advanced artificial intelligence approach based on physics-informed neural networks (PINNs) to estimate time-varying parameters from given data for the compartmental model. Our proposed PINNs approach captures the complex dynamics of COVID-19 by integrating a modified Susceptible-Exposed-Infectious-Recovered-Death (SEIRD) compartmental model with deep neural networks. The experimental findings on synthesized data have demonstrated that our method robustly and accurately learns the dynamics and forecasts future states. Moreover, as more data becomes available, our proposed PINNs approach can be successfully extended to other regions and infectious diseases.

## CCS CONCEPTS

• **Computer systems organization** → **Embedded systems**; *Redundancy*; Robotics; • **Networks** → Network reliability.

## KEYWORDS

Compartmental models, COVID-19 transmission, Physics-informed neural networks, Forward-inverse problem

**ACM Reference Format:**
Xiao Ning, Yongyue Wei, and Feng Chen. 2023. Physics-informed neural networks integrating compartmental model for analyzing COVID-19 transmission dynamics. In *Proceedings of Make sure to enter the correct conference title from your rights confirmation emai (Conference acronym 'XX)*. ACM, New York, NY, USA, 8 pages. https://doi.org/XXXXXXX.XXXXXXX

---

*corresponding author

---

## 1 INTRODUCTION

The emergence of severe acute respiratory syndrome coronavirus 2 (SARS-CoV-2) has presented an unprecedented and complex public health challenge, with emerging and re-emerging infectious diseases posing a significant threat. Compartmental models, governed by a nonlinear system of ordinary differential equations (ODEs), simulate multi-state population transitions by incorporating domain knowledge and mathematical assumptions to characterize the transmission dynamics of infectious diseases. These models are a powerful tool for detecting, understanding, and combating infectious disease outbreaks and have been widely used to evaluate the impact of various public health interventions during the COVID-19 pandemic [24]. However, since real-world data can be inherently stochastic, noisy, and even inaccessible, model optimization and methodological innovation are urgently needed to handle imperfect data and provide early warning of major public health emergencies.

Modeling and predicting the behavior of infectious diseases is crucial for early warning and evaluating effective interventions to mitigate damage. The first compartmental model, Susceptible-Infectious-Removed (SIR), was proposed by Kermack and McKendrick to study the epidemics of the Black Death in London and the plague in Mumbai [12]. Compartmental models allow the addition of compartments or transmission parameters to explore and estimate the impact of different assumptions regarding interventions. These parameters, included in the compartmental model, determine the transmission progress between different disease statuses and can generate essential characteristics of an epidemic [2]. Finding the best-fit parameters from the system, given available data, is an inverse problem. Several numerical methods have been developed to infer constant model parameters from available data. These methods convert the inverse problem into an optimization problem and formulate an estimator by minimizing an objective function. However, since various non-pharmaceutical interventions (NPIs) are employed during the evolution of COVID-19, some model parameters are time-varying.

Identifying time-varying parameters in compartmental models is a complex inverse problem, making it challenging to accurately model outbreak dynamics [1, 10]. Recent advances in Physics-informed machine learning have shown promise in COVID-19 transmission modelling by incorporating prior knowledge into deep neural networks to enhance their accuracy and robustness [11]. For

example, Kharazmi et al. used PINNs to identify time-dependent parameters and data-driven fractional differential operators in several epidemiological models [13]. Long et al. proposed a variant of PINNs to fit daily reported cases and identify time-varying parameters in the susceptible-infectious-recovered-deceased model for the spread of COVID-19 [15]. Nascimento et al. introduced an approach that combines physics-informed and data-driven kernels to reduce the gap between predictions and observations [17]. Cai et al. employed fractional physics-informed neural networks to refine the classical susceptible–exposed–infected–removed (SEIR) model, infer time-dependent parameters, and identify unobserved dynamics of the fractional SEIR model [3]. However, most of these approaches only consider the transmission rate as a function of time, while setting other parameters to fixed values. Additionally, they mainly use time-varying parameters for prediction and lack a systematic epidemiological analysis.

The primary focus of this paper is to introduce a novel method for evaluating time-varying parameters in ODEs-based compartmental models and to assess the impact of the NPIs based on the estimated parameters. We constructed a SEIRD compartmental model that takes an incubation period and the corresponding infectivity into account, including both unknown time-varying and constant parameters. Given many unknown parameters and limited data, we modeled the system of ODEs as one network and the time-varying parameters as another network to reduce the parameter of neural networks. Furthermore, such structure of the PINNs approach is in line with the prior epidemiological correlations. We then tested the effectiveness of our methodology using real-world reported data, simulation experiments showed that our proposed PINNs method effectively performs data-driven parameter estimation for modelling COVID-19 transmission. Moreover, as more data becomes available, it can be successfully extended to model and analyze infectious disease transmission dynamics in various regions and for different infectious diseases.

## 2 METHODOLOGY

### 2.1 Compartmental model

Compartmental models enable the simulation of multi-state population transitions by incorporating domain knowledge and mathematical assumptions to characterize the dynamics of infectious diseases. These models are generally represented as the following nonlinear dynamical system:

$$\begin{cases} \dfrac{d\boldsymbol{U}(t)}{dt} = \mathcal{F}(t, \boldsymbol{U}(t); \Xi) \\ \boldsymbol{U}(t_0) = U_0 \end{cases} \quad (1)$$

where $\boldsymbol{U}(t) \in \mathbb{R}^D$ (typically $D \gg 1$) is the state variable, $\boldsymbol{t} \in [t_0, T]$ is the time range, $U(t_0)$ is the initial state, and $\Xi$ stands for the parameters of the dynamical system.

The SIR compartmental model provided the simplest framework that matched the reporting structure with the least underlying assumptions. Many variations of the SIR model have been proposed to analyze the transmission of COVID-19. In this paper, we consider a geographical region as isolated from other regions, and within such region we divide the population ($N$) of study region into five compartments, susceptible ($S$, vulnerable to COVID-19 infection),

exposed ($E$, latent individual or asymptomatic infective), infected ($I$, symptomatic infected), recovered ($R$, immune to COVID-19), and dead ($D$, death due to COVID-19). The details of the SEIRD model are described below:

$$\begin{cases} \dfrac{dS(t)}{dt} = -\dfrac{\beta S(t)(\epsilon E(t)) + I(t)}{N} \\ \dfrac{dE(t)}{dt} = \dfrac{\beta S(t)(\epsilon E(t) + I(t))}{N} - \dfrac{E(t)}{\alpha} \\ \dfrac{dI(t)}{dt} = \dfrac{E(t)}{\alpha} - \gamma I(t) - \mu I(t) \\ \dfrac{dR(t)}{dt} = \gamma I(t) \\ \dfrac{dD(t)}{dt} = \mu I(t) \\ \quad N = S(t) + E(t) + I(t) + R(t) + D(t) \end{cases} \quad (2)$$

Where $S(t), E(t), I(t), R(t), D(t)$ denote the number of susceptible, exposed, infectious, recovered, and deceased individuals over time respectively, along with non-negative initial conditions $S(0) = S_0, E(0) = E_0, I(0) = I_0, R(0) = R_0, D(0) = D_0$. $\beta \geq 0$ represents the transmission rate, which represents the probability of infection per exposure when a susceptible individual ($S$) has contact with an infected patient ($I$) and becomes a latent exposed individual ($E$). A coefficient parameter $\epsilon$ is introduced since the transmission capacity of exposed and onset populations may be different. $\epsilon\beta$ represents the potential rate per exposure when a susceptible individual ($S$) has mutual contact with an exposed individual ($E$), and transmits it to another exposed individual ($E$). $\alpha$ is the average duration of incubation period, $1/\alpha$ is the rate of latent individuals becoming infectious Besides, $\gamma \geq 0$ represents the recovery rate, $\mu \geq 0$ represents the death rate, and $N$ is the total population.

The assumption that the parameters in Eqs. 2 are time-constant, which is a highly restrictive and unrealistic one for the real-world epidemic where various interventions exist. The associated interventions implemented by authorities, and/or mutations of the virus, et al. make the compartmental model require time-varying parameters to capture the dynamic of dynamics of COVID-19. Therefore, by considering transmission rate $\beta$, recovery rate $\gamma$ and death rate $\mu$ as functions of time $\beta(t), \gamma(t), \mu(t)$, the re-written SEIRD model is as follows:

$$\begin{cases} \dfrac{dS(t)}{dt} = -\dfrac{\beta(t)S(t)(\epsilon E(t)) + I(t)}{N} \\ \dfrac{dE(t)}{dt} = \dfrac{\beta(t)S(t)(\epsilon E(t)) + I(t))}{N} - \dfrac{E(t)}{\alpha} \\ \dfrac{dI(t)}{dt} = \dfrac{E(t)}{\alpha} - \gamma(t)I(t) - \mu(t)I(t) \\ \dfrac{dR(t)}{dt} = \gamma(t)I(t) \\ \dfrac{dD(t)}{dt} = \mu(t)I(t) \\ \quad N = S(t) + E(t) + I(t) + R(t) + D(t) \end{cases} \quad (3)$$

Among them, the five variables $S(t), E(t), I(t), R(t), D(t)$ have the same meanings as in Eq. 2. If we assume that the total population $N$ is constant, then the sum of the increase or decrease of the state of each population is 0, namely, $\frac{dS(t)}{dt} + \frac{dI(t)}{dt} + \frac{dR(t)}{dt} + \frac{dD(t)}{dt} = 0$.

The basic reproduction number $R_0$ is a constant epidemiological parameter that provides an estimation of the contagiousness of the infectious disease. It also serves as a threshold parameter, when $R_0 > 1$, one infected individual can trigger an outbreak, while when $R_0 < 1$, the infection will not spread in the population. Given a compartmental model, $R_0$ can be calculated by the Next Generation Matrix (NGM) approach [7].

If the related parameters in the compartmental model are time-varying as in Eq. 3, the reproduction number $R_0$ is expected to keep changing, as a function of time called the effective reproduction number $R_t$. $R_t$ for the course of SEIRD model using the NGM approach, which yields the following expressions in the proposed SEIRD model:

$$R_t = \epsilon \cdot \beta(t)\alpha + \frac{\beta(t)}{\gamma(t) + \mu(t)} \tag{4}$$

$R_t$ provides an estimation of the contagiousness of the infectious disease, during the course of an outbreak, where not every individual is considered susceptible.

## 2.2 Deep neural networks

Deep neural networks (DNNs) have emerged as a reliable and effective method for nonlinear function approximation, demonstrating remarkable capabilities in scientific computation and engineering applications, as evidenced by their widespread utilization. Many types of DNNs have been developed such as recurrent neural networks, convolutional neural networks, and Transformer et al [16], and here we only consider fully-connected deep neural networks (FDNN). Neural networks can be viewed as discretizations of continuous dynamical systems, making them well-suited for dealing with dynamic systems. Mathematically, an FDNN defines a mapping of the form

$$\mathcal{F} : x \in \mathbb{R}^d \Longrightarrow y = \mathcal{F}(x) \in \mathbb{R}^c, \tag{5}$$

where $d$ and $c$ are the input and output dimensions, respectively. Generally, a standard neural unit of an FDNN receives an input $x \in \mathbb{R}^d$ and produces an output $y \in \mathbb{R}^m$, $y = \sigma(Wx + b)$ with $W \in \mathbb{R}^{m \times d}$ and $b \in \mathbb{R}^m$ being weight matrix and bias vector, respectively. $\sigma(\cdot)$, which is referred to as the activation function, is designed to add element-wise non-linearity to the model. An FDNN with $\ell$ hidden layers can be considered a nested composition of sequential standard neural units. For convenience, we denote the output of the DNN by $y(x; \theta)$ with $\theta$ standing for the set of all weights and biases. Specifically, the $j_{th}$ neuron in $\ell$ layer can be formulated as

$$y_j^{[\ell]} = \sum_{k=1}^{n^{[\ell-1]}} w_{jk}^{[\ell]} \sigma^{[\ell-1]}(y_k^{[\ell-1]}) + b_j^{[\ell]}, \tag{6}$$

where $y_k^{[\ell-1]}$ represents the value of the $k_{th}$ neuron in the $\ell-1$ layer, $n^{[\ell-1]}$ represents the number of neurons in the $\ell-1$ layer, $\sigma^{[\ell-1]}$ is the activation function of the $\ell-1$ layer, $w_{jk}^{[\ell]}$ is the weight between the $k_{th}$ neuron in the $\ell-1$ layer and the $j_{th}$ neuron in the $\ell$ layer, and $b_j^{[\ell]}$ is the bias of the $j_{th}$ neuron in the $\ell$ layer.

The nonlinear activation function enhances the ability of DNN to model various non-linear problems, selecting the suitable activation function matters greatly for DNN applied in all domains. Particularly, the activation function has an extremely significant

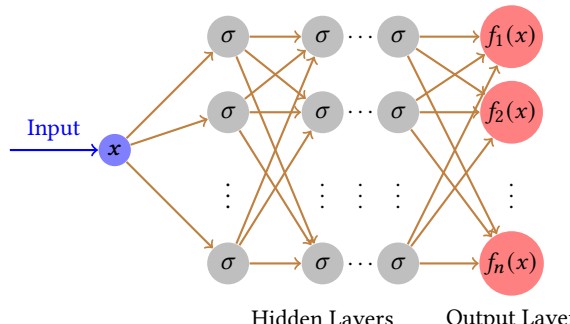

Figure 1: Illustration of the FDNN. A neural network consists of an input layer (the input $x$), several hidden layers (composed of weights $W^\ell$, bias $b^\ell$, and activation function $\sigma$), and an output layer.

impact on the success of training PINNs. *ReLU* activation function has been widely used in many deep learning applications due to its dealing well with vanishing gradients problems [19]. However, for solving differential equations, the first and second derivatives of the neural networks would serve as inputs to calculate the loss function, which means that the activation function of the DNN in PINNs framework requires the second derivative to be satisfied as non-zero. Definitely, many research works have demonstrated that *sigmoid* function and *tanh* function are suited for effective PINNs framework training tasks.

## 2.3 PINNs for SEIRD model

Physics-informed neural networks (PINNs) approach is a data-driven approach to approximate the solution of differential equations and estimate unknown parameters. The main idea of PINNs is to integrate a priori knowledge as physical laws or domain expertise modelled by differential equations into deep neural networks. Equations in the compartmental model possess coupling and the coefficients are not independent of each other through the lens of biological and epidemics. In this context, we employ two separate DNNs with input $t$ to represent the stats $U(t)$ and time-varying parameters, respectively. For the two unknown constant parameters $(\alpha, \epsilon)$, we designed the modified *tanh* activation function to represent them. The expression of the *tanh* function is $tanh(x) = \frac{e^x - e^{-x}}{e^x + e^{-x}}$, and the range of values belong to [-1, 1]. Considering that $\alpha > 0$ and $0 \leq \epsilon \leq 1$, thus we designed the expression of $\epsilon$ as $tanh(x)$, the expression of $\alpha$ as $21 \cdot tanh(x)$, $x$ is a random sample with uniform distribution generated from the interval [0, 3]. Meanwhile, COVID-19 transmission involves the analysis of real-world data, for which the available data size tends to be small and sparse. Such a PINNs architecture enables a well-trained model with a limited data set.

The PINNs framework is required to fit the data and simultaneously satisfy the equations, whereby the loss function includes two parts. The first part is the mismatch between the network output and the available data, and another part is the residual of ODEs. In this study, we employ the approximation $U_{NN}(t; \Theta_U) \approx U(t)$ to

represent the time-varying SEIRD equations (Eqs 3). The parameters $\Theta$ are optimized to achieve the best fit with the observed data. Considering the available data $U_j$ at times $t_1, t_2, ..., t_n$ as training points (ground truth), the mean squared error (MSE) is calculated as follows:

$$MSE_u = \frac{1}{N} \sum_{j=1}^{N} \left| \hat{U}_{NN}(t_j) - U(t_j) \right|^2, \tag{7}$$

Another component of the loss function is the residual of the systems of Eqs. 1, we define the residual of equations as $\mathcal{R}_{NN}(t) = \frac{dU(t)}{dt} - \mathcal{F}(U_{NN}, t; \Xi)$. The residual, denoted as $R(t; \Theta U)$, serves as a metric for assessing the accuracy of the approximation $UNN(t; \Theta_U)$ in satisfying the ordinary differential equations (ODEs). Evaluating the residual involves computing the time derivative of the neural network output, which can be accomplished using automatic differentiation [20]. Automatic differentiation is a computational technique that efficiently computes derivatives by applying the chain rule. It breaks down functions into elementary operations and calculates their derivatives, allowing for accurate and efficient computation of the overall function's derivative with respect to its input variables.

$$MSE_r = \frac{1}{N} \sum_{j=1}^{N} \left| \mathcal{R}_{NN}(t_j) \right|^2, \tag{8}$$

In summary, the loss function of proposed PINNs approach is defined as:

$$L = \omega_u MSE_u + \omega_r MSE_r \tag{9}$$

The weight coefficients, $\omega_u, \omega_r$, in the loss function play a crucial role in balancing the optimization process between learning from the data and satisfying the ODEs. These parameters allow fine-tuning of the model's behaviour and trade-off between the two objectives. By adjusting the values of $\omega_u, \omega_r$, the emphasis can be placed on either accurately fitting the available data or ensuring the ODE constraints are well-satisfied.

Consequently, this PINNs model strives to minimize the loss function, effectively learning the underlying physics encoded in the ODEs while accurately capturing the patterns and relationships in the available data.

## 3 EXPERIMENTS

In this section, we will provide a description of the collected data and present the results obtained from parameter estimation and predictions using the proposed PINNs approach.

### 3.1 Data source

For the COVID-19 epidemic in Italy, the first official report of indigenous case was on February 21, 2020 in Lodi province, while several epidemiological-linked cases were traced back to February 20, 2020. The data considered in our study is downloaded from Italian Civil Protection (http://www.protezionecivile.gov.it/media-comunicazione/comunicati-stampa) and Ministry of Health (http://www.salute.gov.it/portale/home.html). It is comprised of commutative infected, recovered, and deceased cases for the period from February 20, 2020 (day 1), to June 30, 2020 (day 132) [8]. To avoid weekly fluctuations induced by the work-leisure shift and nature noise in the real-world data, a 7-day

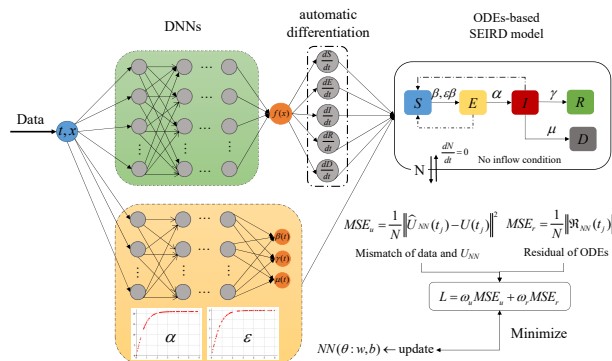

**Figure 2: Schematic diagram of the PINNs framework for the SEIRD compartmental model with unknown (time-varying and constant) parameters. The green-shaded DNNs represents the states $U_{NN}(t)$ to fit the available data and infer the unobserved dynamics. The yellow-shaded DNNs represents time-varying parameters $\beta(t), \gamma(t), \mu(t)$. The two constant parameters $(\alpha, \epsilon)$ are represented by the modified $tanh(t)$ activation function.**

moving average was used to smooth the reported data by averaging the values of each day with those of the 7 days before. In order to control the transmission of COVID-19 in Italy, lockdown and many restriction measures were implemented from February 23, 2020, as the developed timeline shown in Fig. 3. All events and interventions are available from official websites https://mn.gov/governor/covid-19/news/.

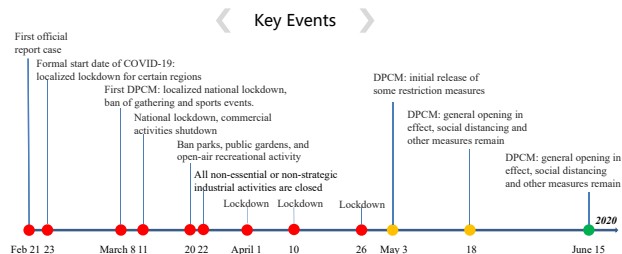

**Figure 3: Timeline of NPIs implemented in Italy to control COVID-19. DPCM: Decree of the Prime Minister.**

### 3.2 Experimental settings

We train the PINNs model on a personal laptop running the Windows 10 operating system, equipped with an Intel (R) Core (TM) i7-8550U CPU operating at 1.8GHz. We implement the PINNs approach using Python and the PyTorch framework [21]. For the numerical experiment, we train the neural networks using the Adam optimizer with an initial learning rate of $2 \times 10^{-3}$ and a decay rate of 95% every 2000 epochs. The entire training process takes about 10 minutes to run 50,000 epochs on all training data, and predictions can be made within seconds.

## 3.3 Results

*3.3.1 Data fitting.* In this subsection, we present the evaluation of how well the estimated parameters fit the SEIRD compartmental model on the available data. Fig.4 shows the fitting of the dynamic of the SEIRD model to the available real-world reported data (after 7-day smoothing), which demonstrates that the proposed PINNs approach can accurately fit the different fluctuations in the data.

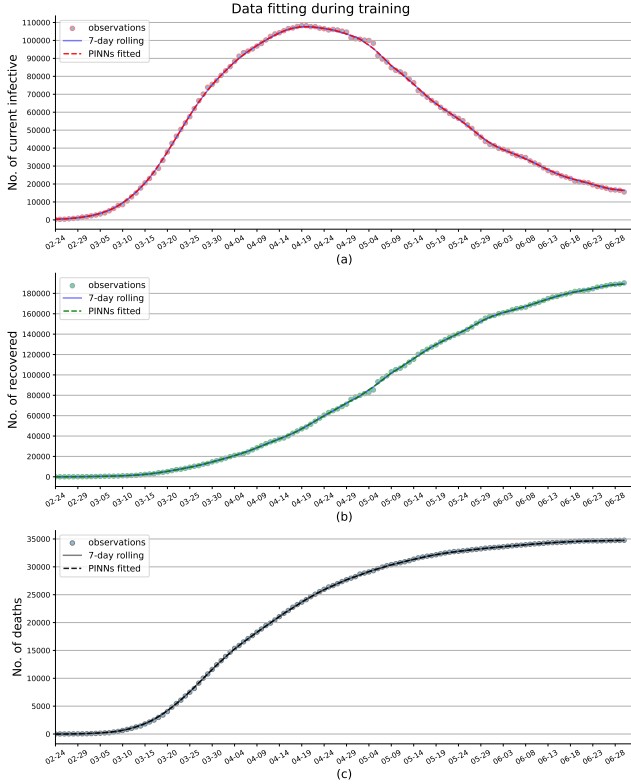

**Figure 4: Data fitting during training. (a.) Fitting to the available data of current infectious. (b.) Fitting to the available data of cumulative recovered. (c.) Fitting to the available data of cumulative deaths. Dot: observed data. Line: 7-day rolling of observed data. Dashed: PINNs' prediction of dynamics.**

*3.3.2 Inference.* We aim to infer the time-varying parameters $\beta(t)$, $\gamma(t)$, $\mu(t)$, as well as the constants $\alpha$ and $\epsilon$, through the inverse problem solving of the SEIRD compartmental model. The incubation period and the infectiousness during this period are parameters specific to the virus, which can be obtained from clinical case information or inferred using statistical or mathematical modelling based on available data. In our study, we estimate the incubation period of COVID-19 to be approximately 5.8 days, and the infectiousness during the incubation period is found to be nearly equal to 99.9% of the infection period.

The transmission dynamics of infectious diseases are influenced by multiple factors, such as government interventions, individual behaviour, and medical resources. In order to accurately model

the spread of infectious diseases using compartmental models, it is necessary to update certain parameters over time to account for the evolving impact of interventions. These parameters include $\beta(t)$, $\gamma(t)$, and $\mu(t)$, which represent the time-varying rates of transmission, recovery, and mortality, respectively. In Figure 5, we present the inference results of these time-varying parameters in Italy from February 20 to June 30, 2020. This analysis provides insights into how the values of $\beta(t)$, $\gamma(t)$, and $\mu(t)$ change over the specified time period, reflecting the impact of interventions and other factors on the dynamics of the disease.

Note that the events that have an impact on $\beta(t)$ have to do with people's adaption to preventive interventions and the interactions among individuals, whereas $\mu(t)$ relates to the availability and effectiveness of health care, as well as on the resource availability in hospitals. $\gamma(t)$ is known to be a disease-specific parameter (inverse of the infectious period) but is also affected by the capacity of the healthcare system to accommodate hospitalization. As shown in Fig.5 (a), the transmission rate $\beta(t)$ can fit well with what would be expected given such events. The earliest traceable first confirmed case of COVID-19 on February 20, 2020, the authorities of Italy started imposing the localized lockdown for certain regions on February 23, 2020, these control measures achieved a certain success, as demonstrated by a significant reduction in transmission rates $\beta(t)$. As far as $\gamma(t)$ and $\mu(t)$, hospitals' ability particularly emergency rooms had a considerable impact. In the context of COVID-19, hospitals are at full capacity in the first months of the outbreak, and as months went by, healthcare professionals learned more about possible treatments to treat the disease's symptoms and effects. This usually results in a decrease in the proportion of individuals that died from the disease (decrease of $\mu(t)$) and in a decrease in the recovery time (an increase of $\gamma(t)$). As shown in Fig.5 (b) and Fig.5 (c), in qualitative terms, was an increasing trend in $\gamma(t)$ and a decreasing trend in $\mu(t)$.

The effective reproduction number is a crucial parameter in the SEIRD model that helps to predict the spread of infectious diseases. $R_t$ less than 1 indicates that the transmission of the infectious disease will gradually disappear. By monitoring changes in $R_t$ over time, public health officials can make informed decisions about interventions to control the spread of the disease. Fig. 6 (a) shows the evolution of $R_t = \epsilon \cdot \beta(t)\alpha + \frac{\beta(t)}{\gamma(t)+\mu(t)}$ in the proposed SEIRD compartmental model from February 20 to June 30, 2020. In the first several days of the outbreak, the effective reproduction number $R_t$ was greater than 8, which resulted in a substantial outbreak. On February 25, $R_t$ gradually decreased as localized lockdown for certain regions and the awareness of the epidemic. However, $R_t$ was still greater than 1, which may be due to the partially incomplete lockdown, or the movement of people from northern to southern Italy when the country-wide lockdown was announced but not yet enforced. When the national lockdown was fully operational and strictly enforced, $R_t$ keeps decreasing and finally reached below 1. Moreover, $R_t$ steadily declined at the end of March due to a wider testing campaign that identified more mildly symptomatic infected individuals. Since June 15, $R_t$ shows a growing trend due to DPCM declaring that general opening was in effect, social distancing, and other measures remained. Additionally, to validate the estimated $R_t$, a serial Bayesian model was implemented to produce the $R_t$ of Italy

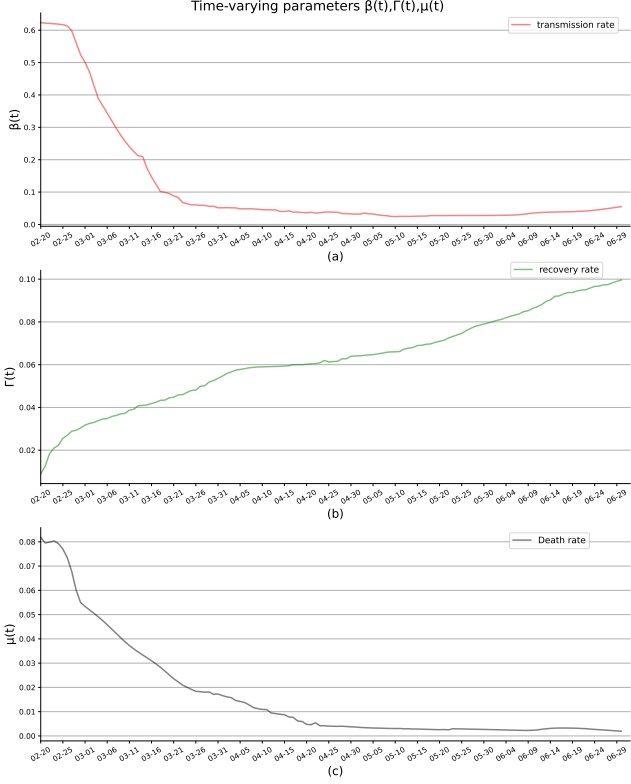

**Figure 5: The time-varying transmission rate of SEIRD model based on PINNs approach on Italy data from February 20 to June 30, 2020. (a): transmission rate $\beta(t)$. (b): recovery rate $\gamma(t)$. (c): death rate $\mu(t)$**

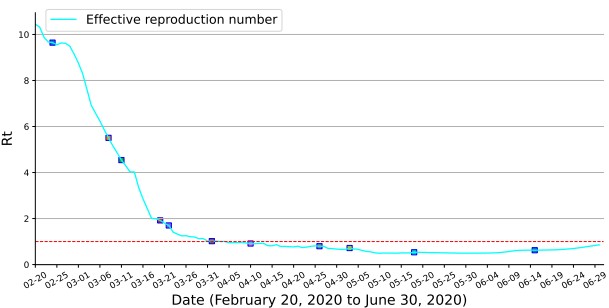

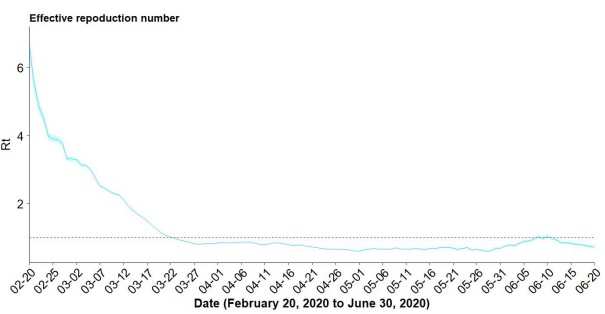

**Figure 6: $R_t$ in Italy from February 24 to June 30, 2020. (a.) Rt estimated by proposed PINNs approach for SEIRD model. (b.) $R_t$ estimated by serial Bayesian model.**

at the same time period [5], as shown in Fig. 6 (b). Parameters for the serial interval distribution in the model were set according to the published literature (mean = 7.5 d; SD = 3.4 d) [18, 23]. As shown in 6, the $R_t$ estimated by the proposed PINNs approach is essentially the same as that estimated by the Bayesian model. Besides, the result of the proposed approach provides a more detailed and accurate capture of the dynamics.

*3.3.3 Forecasting.* Modeling results can provide reliable feedback information for the authorities to make future decisions. The ODEs-based compartmental model requires determined initial conditions and model parameters to make predictions. To test the performance of the proposed PINNs approach, we performed predictions for the early outbreak of COVID-19 in Italy at one-month, two-month, and three-month, respectively. As the initial conditions can be obtained from the training data and the model parameters are already calibrated, we can forecast the epidemic dynamics by performing the forward process. In the prediction part, the value of $\beta(t)$, $\gamma(t)$ and $\mu(t)$ are assumed to be their final value of the training time window. Fig. 7 displays the one-week prediction and corresponding observations for three time periods produced by using the SEIRD model with the estimated parameters. Note that the number of recovered and death states in the SEIRD model are terminal states, which

means that the changes in the number of recovered and death people are always non-decreasing. In turn, the infected people may see periods of increase and decrease due to it being a state of transition. Fig.7 (a) displays the one-week prediction based on the reported data from February 20 to March 20, 2020, Fig.7 (b) displays the one-week prediction based on the reported data from February 20 to April 19, 2020, and Fig.7 (c) displays the one-week prediction based on the reported data from February 20 to May 19, 2020. The perfect match between the predictions and the observations demonstrates the parameters inferred by the learned network are very plausible, as well as the generalization ability of the model.

Furthermore, to quantitatively test the prediction performance of the proposed approach, We use three evaluation metrics to make fair and effective comparisons. They are mean absolute error (MAE), root mean square error (RMSE), and mean absolute percentage error (MAPE). The calculation method is shown in Eq. (10)(12)(11).

$$MAE = \frac{1}{n}\sum_{i=1}^{n}|\hat{y}_i - y_i|, \qquad (10)$$

$$RMSE = \sqrt{\frac{1}{n}\sum_{i=1}^{n}(\hat{y}_i - y_i)^2}, \qquad (11)$$

$$MAPE = \frac{1}{n}\sum_{i=1}^{n}\frac{|\hat{y}_i - y_i|}{\hat{y}_i} * 100\%, \qquad (12)$$

Interventions to control COVID-19 keep adjusting, which may result in uncertainty, experimental results as represented in Table1 show the highly accurate forecasting capability of the proposed approach.

**Table 1: The forecasting performance in 3-day, 5-day and 7-day.**

| Metrics | After March 20, 2020 | | | After April 19, 2020 | | | After May 19, 2020 | | |
|---|---|---|---|---|---|---|---|---|---|
| | 3-day | 5-day | 7-day | 3-day | 5-day | 7-day | 3-day | 5-day | 7-day |
| MAE(I) | 5411 | 5790 | 6419 | 2503 | 3258 | 2792 | 1352 | 2170 | 3046 |
| RMSE(I) | 5431 | 5819 | 6519 | 3705 | 2618 | 3275 | 1567 | 2515 | 3514 |
| MAPE(I) | 11.60% | 11.52% | 11.78% | 2.32% | 3.04% | 2.61% | 2.20% | 3.70% | 5.41% |
| MAE(R) | 813 | 1728 | 2944 | 2934 | 5704 | 9001 | 1643 | 2700 | 4170 |
| RMSE(R) | 959 | 2128 | 3706 | 3321 | 6821 | 10936 | 1880 | 3151 | 4972 |
| MAPE(R) | 11.93% | 20.07% | 31.04% | 5.57% | 10.00% | 14.83% | 1.23% | 1.96% | 2.97% |
| MAE(D) | 423 | 543 | 927 | 330 | 235 | 318 | 147 | 109 | 95 |
| RMSE(D) | 527 | 637 | 1151 | 349 | 279 | 379 | 147 | 122 | 109 |
| MAPE(D) | 8.36% | 8.98% | 12.64% | 1.35% | 0.95% | 1.24% | 0.45% | 0.34% | 0.30% |

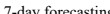

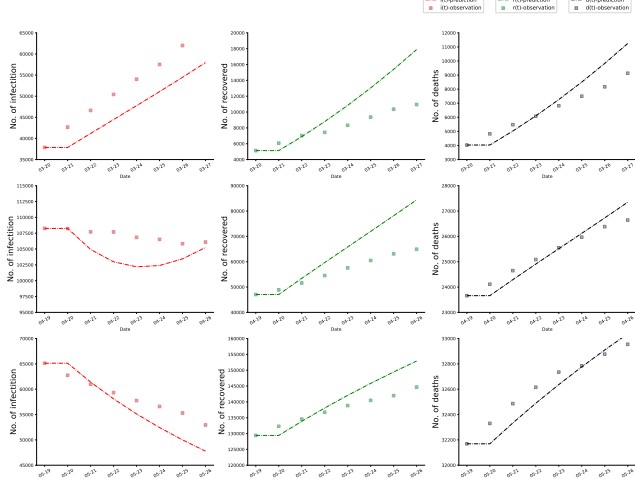

**Figure 7: Forecasting results of the SEIRD models based on estimated parameters. In the first column are plotted the predicted current infections, in the second column are plotted the predicted cumulative recovered, in the third column are plotted the predicted cumulative deaths, and the dotted boxes represent the corresponding observations. a. 7-day forecasting results based on the February 20 to March 20, 2020 time window. b. 7-day forecasting results based on the February 20 to April 19, 2020 time window. c. 7-day forecasting results based on the February 20 to May 19, 2020 time window.**

## 4 DISCUSSION

Transmission modelling is increasingly being used to support public health decision-making in the control of infectious diseases. In this paper, a modified SEIRD compartmental model with time-varying parameters is introduced to describe and predict the dynamics of COVID-19 transmission in Italy.

Estimating the unknown parameters of this model is a complex inverse problem, for the solution of which we proposed a domain-specific PINNs approach.

The proposed approach has been applied to modelling the COVID-19 transmission in Italy, the estimated parameters resulted effective in fitting the COVID-19 contagion data and in providing accurate predictions of the evolution. Besides, these results, the proposed PINNs approach allows us to have a more detailed understanding of the contagion mechanism.

In Fig. 5 (a) is that the control measures imposed by the authorities seem to have been effective in reducing the key transmission rate parameter $\beta(t)$. Fig. 5 (b) and (c) show that the recovery rate tends to increase with time and the death rate to decrease. This phenomenon, which seems not directly related to the lockdown, can be attributed to different causes, among which a better understanding of the disease and consequent improvement in the effusiveness of the response from the national health system, and possibly a change in the nature, virulence, and lethality of the virus. Furthermore, we evaluate how the estimated parameters fit the SEIRD compartmental model by comparing the results of previous publications. We compare our results to those obtained using the methodology of the rolling regression framework [4], where the order of magnitude of the time-varying parameters $\beta(t)$, $\gamma(t)$ and $\mu(t)$ is in agreement and the trend is almost identical. A comprehensive meta-analysis demonstrated that the median incubation period for general transmissions in early outbreaks was 5.8 days [95% confidence interval (95% CI): 5.3, 6.2] [25]. Li et al. analyzed data on the first 425 confirmed cases in Wuhan to determine the epidemiologic characteristics of NCIP, the results show that the mean incubation period was 5.2 days (95% confidence interval [CI], 4.1 to 7.0) [14]. Yang et al. collected contact tracing data in a municipality in Hubei province during a full outbreak period to estimate the incubation period and serial interval of COVID-19, the estimated median incubation period of COVID-19 is 5.4 days (bootstrapped 95% confidence interval (CI) 4.8–6.0) [26]. The estimated $\alpha$ by the proposed PINNs approach is 5.8, which is consistent with the results of the above research. The estimated $\epsilon$ by the proposed PINNs approach is 0.99, which means that the transmission capacity of exposed and onset populations are nearly identical [9]. Numerous related studies demonstrate that the incubation period and the infection period carry almost the same capacity for transmission [6, 22].

The goal of modeling the transmission dynamics of an infectious disease is to capture the mechanisms of a host passing on the infection to other individuals. Once the information is clear, a model can be used as a sort of experimental system to simulate what would happen to the evolution of disease with different interventions implemented. While the proposed PINNs approach indeed offers many advantages, it does have some limitations. One of the main limitations is that PINNs architecture requires prior knowledge of the physical laws and constraints that govern the problem being solved. The structure of compartmental models may change depending on the question of interest and impact their accuracy. That means if the underlying epidemiological laws are not well understood or if the available data is not consistent with the known epidemiological laws, the model may not work well. But it should be noted that the emphasis on infectious disease models is on the application of public health, not the mathematics of these models. As world-renowned Statistician George E. P. Box made the following statement. "All models are wrong, but some are useful."

## 5 CONCLUSIONS

In this paper, we proposed a novel PINNs approach to estimate the unknown parameters (including time-varying and constant parameters) for the ODEs-based compartmental model to depict the dynamic of the COVID-19 transmission. The experiment result with real-world report data reveals that the proposed COVID-19 modeling approach enables to yield of epidemiological models that can describe the real-time dynamics of the contagion, providing reliable predictions and valuable insight into the contagion mechanisms. We have provided a completed workflow for analyzing infectious disease transmission systems described by a system of ODEs produced compartmental model. We emphasize that the proposed PINNs approach can easily be implemented without any background knowledge about numerical analysis (for example, stability conditions) but about some libraries for implementing neural networks. For a given scenario that we consider, the proposed PINNs approach can be effective for simulating different epidemic scenarios, testing various hypotheses, and for designing suitable control measures.

## 6 ACKNOWLEDGMENTS

The study was supported by the National Natural Science Foundation of China (82041024 to Feng Chen and 81973142 to Yongyue Wei). This study was also partially supported by the Bill & Melinda Gates Foundation (INV-006371).

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
