# OpenReview forum: "Physics-informed neural networks integrating compartmental model for analyzing COVID-19 transmission dynamics"
_KDD.org/2023/Workshop/epiDAMIK — KDD 2023 Workshop epiDAMIK_

### Official Review · Reviewer_xqZi · 2023-06-15
**Interesting but not significant**

**Rating:** 1
**Confidence:** 4

**Review:**

Review Summary:

This paper presents an approach utilizing physics-informed neural networks (PINNs) to estimate time-varying parameters in compartmental models for infectious diseases. The authors successfully integrate the SEIRD model with deep neural networks to capture the dynamics of COVID-19, demonstrating proficient learning and accurate future state predictions using the PINNs approach. The results showcase the potential applicability of this method to various regions and infectious diseases. Nonetheless, the absence of comparative analysis with existing methods and the suboptimal forecasting performance depicted in Figure 7 and Table 1 raise notable concerns. Comparable studies (references [1] and [2] which are both compartmental model + deep neural networks for COVID-19 dynamics) have achieved superior performance with simpler compartmental models, specifically the SIRD model instead of the SEIRD model. Additional empirical evidence or theoretical support is imperative to substantiate the significance of this work.

Pros:
1. Introduction of an advanced artificial intelligence approach based on physics-informed neural networks for estimating time-varying parameters in compartmental models.
2. Integration of the SEIRD model with deep neural networks to capture the complex dynamics of COVID-19.
3. The potential applicability of the proposed approach to other regions and infectious diseases.

Cons:
1. Lack of comparison with existing methods and inadequate rationale behind the proposed model.
2. Suboptimal performance was observed in the forecasting results presented in Figure 7 and Table 1.
3. Similar studies (e.g., references [1] and [2]) have achieved superior performance.
For instance, in [1], the mean absolute error (MAE) of parameter I for 3-day forecasting is reported as 29.57. In [2], the MAE of I for 3-day forecasting is documented as 251.73 and 200.24. Conversely, in this work, the MAE of I for 3-day forecasting is significantly larger, ranging from 5411 to 1352.

Without substantial empirical evidence or theoretical support to establish the significance of this work, I am inclined to believe that its quality and significance may not meet the criteria for acceptance.

[1] Ning, Xiao, et al. "Epi-DNNs: Epidemiological priors informed deep neural networks for modeling COVID-19 dynamics." Computers in biology and medicine 158 (2023): 106693.

[2] Ning, Xiao, et al. "Euler iteration augmented physics-informed neural networks for time-varying parameter estimation of the epidemic compartmental model." Frontiers in Physics 10 (2022): 1300.

---

### Official Review · Reviewer_eTpJ · 2023-06-27

**Rating:** 3
**Confidence:** 4

**Review:**

### Summary

This work studies using physics-informed neural networks to estimate the unknown parameters of epidemic compartmental models. To achieve this, this work first proposes an extended counterpart model, named SEIRD, to model the dynamics of the COVID-19 pandemic, which takes the Death (D) counts into consideration. This paper then posits that the parameters for the pandemic in different phases are dynamic. Therefore, this paper proposes a graph neural network to fit the reported cases and epidemic model parameters simultaneously. In experiments, this work evaluates the proposed GNN on the reported cases from Italy. The results show that the model is able to fit the reported cases and generate corresponding epidemic model parameters. Lastly, the model is applied to forecast the infected cases. The results show that the model predicts the reported cases reasonably accurately, mostly achieving within 20% relative absolute error.

### Strengths

- This paper designs a physics-informed neural network to fit case counts and the parameters  in the epidemic model simultaneously and considers the dynamics of the epidemic model parameters.
- The empirical study on the reported cases from Italy shows that model fits the reported cases accurately and generates meaningful model parameters.

### Weaknesses

- It would be interesting to connect the intepretation of estimated epidemic parameters to the intervention policicies. Figure 3 shows the intervention policies conducted by the government during the pandemic. I wondering whether the estimated parameters can be incorporated to explain the effect of each intervention policy. Can the local changes of estimated parameters be interpreted corresponding to the application of the intervention policy.
- The proposed methods needs to be described in more details. For example, in Figure 2, there is a automatic differentiation step to convert the estimated cases to its differentiation. It would be helpful to describe how this step is conducted, since it connects the two conterparts of the neural networks.
- Experimental setup is not well described. For example, in data fitting experiments in Section 3.3.1, it would be better for the reader to interprete the results, if the authors can explain the data splitting for fitting the model to the data. Would different data splitting leads to different results?
- Comparison with related baselines needs to be incorporated. This paper shows the error of the model regarding forecasting the reported cases of the pandemic. However, the comparison with related baselines, such as other PINN or time series prediction methods, would be helpful to assess to the effect of the proposed model.
- Discussion of related work is missing. It would be better to provide a more detailed discussion of previous epidemic models and phisics-informed neural networks.

---

### Official Review · Reviewer_twxy · 2023-06-29
**Interesting Idea That Would Benefit From Better Clarity and Justification**

**Rating:** 2
**Confidence:** 4

**Review:**

This paper proposes the use of physics-informed neural networks (PINNs) to estimate time-varying parameters of ODEs to model transmission dynamics for infectious diseases.

Positives:
+ The idea of modeling the transmission dynamics through a SEIRD model with PINNs is an interesting idea and contribution
+ The authors contain sufficient background on related work in order to present their contribution, and how their method is formed
+ The epidemiological analysis throughout the results is much appreciated and provides a deeper appreciation of many of the results obtained
+ The authors do a great job of contextualizing results with respect to the policies enacted in Italy during the beginning of the global pandemic. This contextualization really helps in understanding the learned trends for the time-varying parameters, and for R_t.

Pieces That Could Be Improved:
- Grammar and writing clarity throughout the manuscript could be improved significantly. For example, the description of the compartmental model in section 2.1 could be significantly improved for clarity, as it is currently difficult to fully understand the different parameters of the model. This is an issue throughout the paper and makes the paper hard to follow
- More information about how evaluation is performed should be provided. As it is written, it is unclear if different data were used for training the models and evaluation (in fact, currently, it seems they are the same data). This could pose an issue with proper validation.
- It is not clear whether the reported MAE, RMSE, and MAPE results are sufficiently strong. It would be beneficial to see more baselines to see if the proposed PINN is actually performing well, such as if traditional NNs (such as recurrent neural networks) that only forecast I, R, D without modeling the ODEs perform worse.
- There should be more ablations to understand if their proposed changes to the model actually result in meaningful changes. For example, is the PINN-based activation functions for alpha and epsilon meaningful? And do the two models for the ODE and the time-varying parameters of the simulation make a difference compared to one shared model? As these points were not well-motivated in the methods, it would be useful to see their importance in the real experimental results.

---

### Official Review · Reviewer_23Zo · 2023-06-29
**This paper proposes a physics-informed neural network PINN to estimate time-varying parameters for SEIRD compartmental models as well as demonstrate learning the complex dynamics of disease and forecasting accurately.**

**Rating:** 4
**Confidence:** 5

**Review:**

The paper is of good quality, clear, and well-written. The authors clearly explained their motivation, addressed the shortcomings of previous methods, and touch upon all the necessary architectures. I would like to bring up a few important points that require attention.

1. I'm a bit confused about the design of the constant variables $\alpha$ and $\epsilon$. In order to obtain a positive value, $\alpha$ is set to a positive multiplication of a hyperbolic tangent function. However, some further clarification would be greatly appreciated.

2. I highly appreciate the way all the methodologies are being explained with proper figures and equations.

3. Based on the data presented in Figure 4, the figures depicting the data fitting during training are in perfect alignment with the observed data points. I am curious if dropouts were utilized in the model and, if not, whether overfitting occurred. It would be greatly appreciated if the authors could provide their insight on this matter.

4. Based on Figure 5, the forecasting accuracy of $I, R, D$ for three different months appears to be relatively consistent when compared to actual observations.

5. One of the pros of this paper is that the authors discussed the main limitation of PINNs and how requirement of prior knowledge could be a constraint while solving problems and potentially may impact accuracy if underlying epidemiological laws are poorly understood or data inconsistencies exist.

This paper on PINNs for infectious diseases is commendable, delivering accurate weekly forecasting results. While other studies have explored physics-informed neural networks in various compartmental models, such as SIR, SIRS, and SEIRM, this research stands out by successfully delivering on its initial claims.